# Effectiveness of interventions aimed at reducing HIV acquisition and transmission among gay and bisexual men who have sex with men (GBMSM) in high income settings: A systematic review

Janey Sewell[1]*, Ibidun Fakoya[2], Fiona C. Lampe[1], Alison Howarth[1], Andrew Phillips[1], Fiona Burns[1,3], Alison J. Rodger[1,3], Valentina Cambiano[1]

1 Institute for Global Health, University College London, London, United Kingdom, 2 School of Life Course and Population Sciences, King's College London, London, United Kingdom, 3 Royal Free London NHS Foundation Trust, London, United Kingdom

* j.sewell@ucl.ac.uk

## Abstract

### Introduction

HIV transmission continues among gay and bisexual men who have sex with men (GBMSM), with those who are younger, or recent migrants, or of minority ethnicity or who are gender diverse remaining at increased risk. We aimed to identify and describe recent studies evaluating the effectiveness of HIV prevention interventions for GBMSM in high income countries.

### Methods

We searched ten electronic databases for randomized controlled trials (RCTs), conducted in high income settings, and published since 2013 to update a previous systematic review (Stromdahl et al, 2015). We predefined four outcome measures of interest: 1) HIV incidence 2) STI incidence 3) condomless anal intercourse (CLAI) (or measure of CLAI) and 4) number of sexual partners. We used the National Institute for Health and Care Excellence (UK) Quality Appraisal of Intervention Studies tool to assess the quality of papers included in the review. As the trials contained a range of effect measures (e.g. odds ratio, risk difference) comparing the arms in the RCTs, we converted them into standardized effect sizes (SES) with 95% confidence intervals (CI).

### Results

We identified 39 original papers reporting 37 studies. Five intervention types were identified: one-to-one counselling (15 papers), group interventions (7 papers), online interventions (9 papers), Contingency Management for substance use (2 papers) and Pre-exposure Prophylaxis (PrEP) (6 papers). The quality of the studies was mixed with over a third of studies

**Data Availability Statement:** All relevant data are within the manuscript and its Supporting Information files.

**Funding:** AR, FL, AP (Grant holders). The programme of research was funded by the NIHR under its Programme Grants for Applied Research Programme (Grant Reference Number RP-PG-1212-20006): A comprehensive assessment of the cost-effectiveness of HIV prevention and testing strategies, including HIV self-testing, among men who have sex with men (MSM) in the UK (PANTHEON). https://www.nihr.ac.uk/ The funders had no role in study design, data collection and analysis, decision to publish, or preparation of the manuscript.

**Competing interests:** The authors have declared that no competing interests exist.

rated as high quality and 11% rated as poor quality. There was some evidence that one-to-one counselling, group interventions (4–10 participants per group) and online (individual) interventions could be effective for reducing HIV transmission risk behaviours such as condomless anal intercourse. PrEP was the only intervention that was consistently effective at reducing HIV incidence.

## Conclusions

Our systematic review of the recent evidence that we were able to analyse indicates that PrEP is the most effective intervention for reducing HIV acquisition among GBMSM. Targeted and culturally tailored behavioural interventions for sub-populations of GBMSM vulnerable to HIV infection and other STIs should also be considered, particularly for GBMSM who cannot access or decline to use PrEP.

## Introduction

In 2020 there were approximately 1.5 million new HIV infections globally [1], with just under half occurring in Sub Saharan Africa. However, there remains a substantial HIV epidemic in key populations in high income settings including in gay, bisexual and other men who have sex with men (GBMSM). In 2019 GBMSM accounted for 69% of new HIV diagnoses in the United States of America (USA) [2], 39% in the European Union (EU)/European Economic Area (EEA) [3], and 63% in Australia [4], although incidence in high income settings in GBMSM is now declining [3, 5, 6].

Rapid declines in HIV incidence in cities such as San Francisco [7, 8] and London [9] have been partly attributed to Pre Exposure Prophylaxis (PrEP), the use of ART by HIV negative people before exposure to prevent HIV acquisition. Other factors contributing to declining HIV incidence in GBMSM in high income settings include an increase in the frequency of HIV testing leading to earlier HIV diagnosis, and a shorter time to ART initiation and viral suppression [10]. Nonetheless HIV transmission is still occurring among GBMSM, with specific sub populations of GBMSM in high income countries remaining at increased risk, including those of minority ethnicity, or those who have recently migrated, or those who are gender diverse [6, 11]. For these and other groups of GBMSM whose HIV prevention needs are not being met by current HIV prevention interventions, a clearer understanding of effective HIV prevention interventions is required.

Several previous publications have attempted to systematically review and identify the effectiveness of HIV prevention interventions among GBMSM or specific sub populations of GBMSM [12–16]. A systematic review conducted in 2012 to 13 by Stromdahl et al. provided a comprehensive breakdown of the efficacy and effectiveness of a broad range of HIV prevention interventions implemented in a European setting, not restricted to RCTs [16]. Twenty-four HIV prevention interventions for GBMSM were included in Stromdahl's review, eight of which were "strongly" (condom use, universal coverage of ART or treatment as prevention, peer-led group interventions, peer-outreach) or "probably" (voluntary counselling and testing, condom-compatible lubricant use, post-exposure prophylaxis, individual counselling for GBMSM living with HIV) recommended for implementation in Europe, however Stromdahl et al's review was conducted before any data on PrEP effectiveness had been published, as the searches were run in 2012–2013 [16].

The aim of this systematic review is to identify and describe randomised controlled trials evaluating the efficacy or effectiveness of HIV prevention interventions for reducing HIV incidence among GBMSM in high income countries published since Stromdahl et al. in 2013 [16]. In contrast to Stromdahl et al's review we restricted the studies included in this review to randomised controlled trials as these are the gold standard to establish the effectiveness of an intervention, as confounding is minimised due to randomisation.

## Methods

We searched ten electronic databases (PubMed, Embase, Medline, Cinahl, PsycINFO, The Cochrane Library, WHO publication database, Social Policy and Practice, EPPI Centre, Web of Science) in April 2021. Search terms limited eligible studies to those published from 1st January 2013 up until the end of February 2021 in order to capture those not included in the Stromdahl et al. review (for which searches had been performed from December 2012 to February 2013 [16]). To find all HIV-related randomized controlled trials (RCTs) conducted among GBMSM, MeSH terms and key word synonyms for "HIV", "Homosexuality", "Bisexuality" were combined with synonyms for "randomized controlled trial", "placebo" and "drug therapy". Details of the search strategy can be found in S1 Appendix.

### Selection criteria

Only peer-reviewed studies published in English, conducted in countries listed as high income on the World Bank list of economies (see S2 Appendix), and which collected data from 1996 onwards, were included in this review. Studies were eligible for inclusion if the study population included GBMSM aged $\geq 16$ years and the findings from the study sample of GBMSM were disaggregated from any other population samples included in the study. Studies were eligible for inclusion if they were randomised controlled trials, sufficiently powered ($\geq 15$ participants per trial arm) to assess the efficacy or effectiveness of interventions and initiatives directly aimed at reducing HIV incidence among GBMSM. Only studies assessing at least one of the four outcomes related to HIV transmission were included (irrespective of whether these were the primary outcomes for the trials themselves). The chosen outcomes were: 1) HIV incidence; 2) STI incidence; 3) condomless anal intercourse (CLAI) (or a measure of CLAI or condom-use) and 4) number of sexual partners (or a measure of number of partners). Data at end of the trial follow-up period was extracted for these four outcomes.

### Study selection and quality appraisal

Studies were selected using a two-stage screening approach. Three reviewers independently screened the titles and abstracts of a randomly selected sample of 10% studies (JS, AH, VC). Since a high (>90%) rate of agreement was recorded for the sample, the remaining titles and abstracts were screened by one reviewer (JS).

Eligible references were selected for full paper screening and assessment by one reviewer (JS) and checked for accuracy by a second (AH/VC), using a checklist devised *a priori* by the author. Studies published after 2013, but based on data collected before 1996, were excluded at the full paper screening stage. We used the National Institute for Health and Care Excellence (UK) Quality Appraisal of Intervention Studies tool to assess the quality of papers included in the review. This tool (derived from Jackson et al., 2006 [17]) necessitates that each study receive a quality rating for both internal and external validity. Internal validity encompasses a range of criteria that establish whether potential sources of bias have been minimised and the extent to which a study establishes a trustworthy cause-and-effect relationship between a treatment and an outcome. External validity assesses the extent to which the study findings are

generalizable to the whole 'source population' (that is, the population the participants were selected from). Each study was rated ('++', '+' or '-') to indicate its quality. Each paper included in the review was quality assessed by two reviewers (JS/AH/VC/FL). All disagreements were resolved through consensus.

Data extraction was undertaken by one reviewer (JS) and checked for accuracy by another (AH/VC). Data were extracted using forms detailing: study setting, population and objectives; data collection period; intervention and control; inclusion and exclusion criteria; recruitment dates and location, method of intervention allocation, relevant outcome measure(s) and follow-up; results (effect size; attrition); author defined strengths and limitations.

## Data synthesis

After data extraction for each paper, studies were grouped post hoc according to intervention type, using the following categories: one-to-one counselling (either in-person or via telephone or text), group in-person interventions (referred to herein as group interventions), individual online interventions (referred to herein as online interventions), Contingency Management (motivational incentives) for substance use, and HIV Pre-exposure Prophylaxis (PrEP). If a study reported multiple intervention categories (i.e. online and group) it was allocated to the category which it primarily indicated as investigating. As the trials contained a range of effect measures (e.g. odds ratio, risk difference) comparing the outcomes between arms in the RCTs, we converted them into standardized effect sizes (SES) with 95% confidence intervals (CI) [18]. These are unitless measures of effect size and so allow the comparison of effect across studies using different outcomes and effect measures. We calculated effect sizes directly for studies that had a continuous outcome variable as the difference in means divided by the pooled standard deviation [19]; for studies in which the primary outcome was binary, the log odds ratios were calculated, before converting them into approximate effect sizes by dividing them by 1.81 [19]. A positive effect size indicates the outcome was more frequent in the intervention than the control group and a negative effect size indicates that the outcome was less frequent in the intervention group. We assessed the size of effect based on Cohen's suggestion that d = 0.2 (or d = -0.2) be considered a 'small' effect size, 0.5 (or -0.5) represents a 'medium' effect size and 0.8 (or -0.8) a 'large' effect size [18]. Data were analysed using Excel (Microsoft, Redmond, Wash). For a small number of studies [20–24], there was insufficient information to calculate an effect size, in which case we indicated as such in the Tables and text.

## Results

The search process returned 10,539 records from all sources, reducing to 6,645 after excluding duplicates. Title and abstract screening excluded a further 6,346; the remaining 299 full text articles were read to assess eligibility. One study was excluded as it was a non-inferiority trial comparing two types of PrEP and the interest was in identifying interventions that were effective [25]. In total, 37 studies presented in 39 papers, fulfilled the inclusion criteria, underwent quality assessment and were included in the final review. See Fig 1 for details of the study selection process, and Tables 1 and 2 for details of the included studies.

The majority of single-country studies were conducted in the USA (n = 30) [20, 21, 23, 24, 26–51], two in the UK (n = 2) [22, 52], and one each in Canada [53], Hong Kong [54] and Taiwan [55]. Two studies reported the results of multi-country RCTs: one study took place in sites in Belgium, France, Germany, Italy, the Netherlands, Poland, Spain and England [56] and the other was undertaken in France and Canada [57–59] (three papers).

Five intervention types were identified: one-to-one counselling (15 papers) [20–22, 24, 26–35, 56]; group interventions (7 papers) [36–41, 53]; online interventions (9 papers) [23, 42–45,

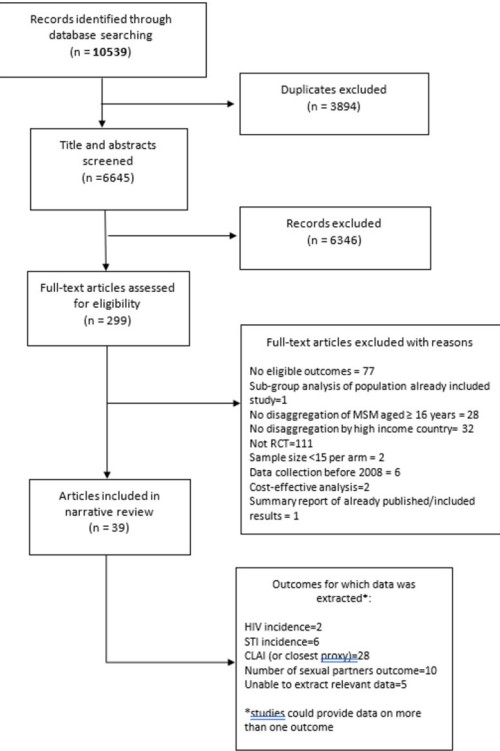

**Fig 1. Flow diagram of papers included at each stage of study selection.**

50, 51, 54, 55]; contingency management for substance abuse (2 papers) [46, 47]; and HIV Pre-exposure Prophylaxis (6 papers) [48, 49, 52, 57–59]. Over a third (35.1%; 13/37) of the studies were rated as high quality and four (10.8%) rated as poor quality RCTs for reasons such as high loss to follow-up, insufficiently powered study, short follow-up time and high refusal rates.

Out of the 39 papers, 34 provided enough information on outcomes to allow calculation of standardised effect sizes. The number of papers reporting the four categories of outcome were as follows: two for HIV incidence [52, 57], six for STI incidence [30, 34, 45, 52, 58, 59], 28 for CLAI (or a measure of CLAI or condom-use) [26–33, 35–44, 46, 48–56] and ten for number of sexual partners [26–28, 36–38, 41, 47, 50, 52]. Overall, one study provided information on all four outcomes [52], none provided information on three outcomes, nine provided on two outcomes [26–28, 30, 36–38, 41, 50] and twenty-four provided data on one of the outcomes [29, 31–35, 39, 40, 42–49, 51, 53–59].

Twenty-one of the 39 papers in this review reported a statistically significant difference between intervention and control for one or more of the trial primary outcomes (see Figs 2–5). There were three studies in which discrepancies were noted between our calculated SES and the original analysis due to different statistical methods such as adjustment for baseline values [28, 33, 36].

## Impact of the interventions on the four chosen outcomes

Forest plots for each of the four chosen outcomes are shown in Figs 2–5. Results in the Figures have been grouped by intervention type as indicated by colour: red is PrEP, blue is one to one counselling, purple is online interventions, green is group interventions, grey is contingency

**Table 1. Study characteristics of randomised controlled trials included in this review, grouped by intervention type.**

| Study (year) (ref) | Country/ Setting | Intervention | Participants | Recruitment | Follow -up | Primary outcome(s) (recall period) | Chosen outcome(s) (at end of follow-up period) for data extraction |
|---|---|---|---|---|---|---|---|
| One-to-one counselling interventions for GBMSM (HIV negative, untested or regardless of HIV status) | | | | | | | |
| Coffin et al (2014) [20] | USA/Public offices | Personalised cognitive counselling (PCC):30 to 50-minute counselling session with rapid HIV test (booster session at 3 months) vs rapid HIV test only | 326 HIV negative or unknown GBMSM (162 intervention, 164 control) | May 2010 – May 2012 | 3 and 6 months | # condomless AI events, # condomless AI partners, # condomless AI events with 3 most recent nonprimary partners (past 3 months) | * |
| Eaton et al (2018) [26] | USA/ Community-based research site | Intervention: single-session (45-min) highlighting misbeliefs using graphic novel & sexual network diagram vs Control: contact-matched, standard or care HIV/STI risk-reduction counselling session (per CDC) | 600 HIV negative GBMSM (300 intervention, 297 control) | Dec 2012— Nov 2014 | 3, 6 and 12 months | Proportion of AI with condoms, # condomless AI acts (insertive & receptive), # male sex partners–(past 3 months) | 1)# condomless AI acts 2)# male sex partners |
| Jemmott et al (2015) [27] | USA/University research centre | Being Responsible for Ourselves (BRO) HIV/STI risk reduction (3 sessions), targeting condom use vs control, targeting physical activity & healthy diet | 595 black GBMSM (295 intervention, 300 control) | Apr 2008— May 2012 | 6 and 12 months | Consistent condom use (0/1 variable: every AI or vaginal intercourse (past 90 days) | 1) Consistent condom use (using a condom every AI or vaginal intercourse in the past 90 days = yes/not using a condom every AI or vaginal intercourse = no) 2) Multiple partners (2 or more partners = yes/ 1 or zero partners = no) |
| Lauby et al (2017) [28] | USA/ Community based organisation | RISE: 6-session (90–120 min) individual-level intervention (conducted by counsellor) vs Control: standard one-session individual-levels HIV risk-reduction intervention (by staff of community partner) | 165 bisexual black men (72 intervention, 93 control) | 2010–2012 | 8 weeks & 5 months (1 week, 3 months after intervention) | # partners; # sexual episodes without condoms (past 3 months) | 1)# condomless AI acts 2)# partners |
| Pachankis et al (2015) [21] | USA/Not stated | ESTEEM: 10-session CBT individually-delivered by psychologist over 3 months vs Control: 3-mth waitlist | 67 HIV negative GBMSM (34 intervention, 33 waitlist control) | 2013 & 2014 | 3 and 6 months | condomless anal sex–(past 90 days) | * |
| Parsons et al (2014) [29] | USA/Research centre | Four individual sessions of MI vs four sessions of content-matched education | 143 young HIV negative or unknown GBMSM (73 intervention, 70 control) | Sep 2007— Aug 2010 | 3 and 12 months | condomless AI with a casual partner; number of days of drug use (past 30 days) | 1)# condomless AI events |
| Crosby et al (2018) [30] | USA/STI clinics | Single-session, clinic based, interactive program delivered via computer Vs standard of care | 600 GBMSM HIV undiagnosed | Sept 2012-Dec 2015 | 4, 8 and 12 months | New chlamydia or gonorrhoea (rectal/ pharyngeal/ anal); consistent use of condoms (past 90 days) | 1)STI incidence 2) Consistent condom use (condoms used in 100% of sexual encounters = yes/ condoms not used in 100%of encounters = no) (imputation analysis, model 6) |

(*Continued*)

**Table 1.** (*Continued*)

| Study (year) (ref) | Country/ Setting | Intervention | Participants | Recruitment | Follow -up | Primary outcome(s) (recall period) | Chosen outcome(s) (at end of follow-up period) for data extraction |
|---|---|---|---|---|---|---|---|
| Llewellyn et al (2019) [22] | UK/STI clinics | 2 telephone sessions of augmented MI, with information and skills building based on the IMB model of behaviour change Vs routine care | 175 GBMSM | May 2011-Dec2012 | 4, 8 and 12 months | UAI, consistent condom use (past 4 months) | * |
| Mimiaga et al (2019) [31] | USA/ Community health clinics/ bars/drug treatment centres | 2 sessions of sexual risk reduction counselling (SRR), ten sessions of behaviour activation with SRR, and one session of relapse prevention; Vs 2 sessions of SRR | 41 HIV Negative GBMSM | Jan 2013-Jan 2015 | 3 and 6 months | CAS with unknown or HIV+ partner, CAS unknown while using meth (past 3 months) | 1)# condomless AI events (with HIV+ or unknown status partner) |
| O'Cleirigh et al (2019) [32] | USA/ Community | 10 individual therapy sessions of CBT-TSC (CBT-trauma and self-care) Vs HIV voluntary counselling and testing | 43 HIV negative GBMSM with a childhood history of sexual abuse | July 2007 – October 2010 | Post treatment, 6 and 9 months | condomless anal/vaginal sex with HIV+ or HIV unknown status partners, psychiatric diagnosis, PTSD symptoms (past 3 months) | 1)# condomless AI events (with HIV+ or unknown status partner) |
| Reback et al (2019) [33] | USA/ Community | 1)interactive text conversations with Peer Health Educators, plus five-times-a-day automated theory-based messages, plus a weekly self-monitoring text-message assessment or, 2) the daily automated messages and weekly self-monitoring assessment or, 3) weekly self-monitoring assessment only | 286 methamphetamine-using GBMSM | March 2014—January 2016 | 2 and 3 and 6 and 9 months | days of methamphetamine use, episodes of sex while under the influence of methamphetamine, and number of episodes of CAI with main male, casual male, anonymous male, and/or exchange male partners (past 30 days) | 1)# condomless AI events (with casual partner) |
| Wray et al (2019) [24] | USA/Online and HIV testing clinics | (1) standard post-test counselling (SPC) alone, or (2) SPC plus Game Plan (GP), a tablet tablet-based BMI for alcohol use and HIV risk. | 40 high risk, heavy drinking GBMSM who sought rapid HIV testing | October-December 2017 | 3 months | # of drinking days, # of binge drinking days, # new anal sex partners, total CAS events (past 30 days) | 1)# condomless AI events 2)# partners |
| **One-to-one counselling interventions for GBMSM (living with diagnosed HIV)** | | | | | | | |
| Nöstlinger et al (2016) [56] | Belgium, Italy, France, Germany, Netherlands, Poland, Spain & England/7 HIV clinics, 1 CBO providing HIV care | CISS (computer-assisted intervention for safer sex): 3 individual 50-min counselling sessions with trained providers using computer-assisted tools vs Control: sexual health advice as part of regular HIV care | 112 HIV positive GBMSM (55 intervention, 57 control) | Feb 2011—Feb 2013 | 3 and 6 months | Condom use at last intercourse, HIV transmission risk score (last sexual intercourse). | 1)Condom use at last sexual intercourse (yes/no) |
| Schwarcz et al (2013) [34] | USA/HIV care settings | Individual PCC session of 1 hr vs routine risk-reduction counselling | 374 HIV positive GBMSM (175 intervention, 196 control) | Nov 2006 – Apr 2010 | 6 and 12 months | # episodes of condomless AI with non-primary male partner of different / unknown serostatus–(past 90 days) | **1)STI incidence (new chlamydia or gonorrhoea diagnosis) |

(*Continued*)

**Table 1.** (Continued)

| Study (year) (ref) | Country/ Setting | Intervention | Participants | Recruitment | Follow -up | Primary outcome(s) (recall period) | Chosen outcome(s) (at end of follow-up period) for data extraction |
|---|---|---|---|---|---|---|---|
| Sikkema et al (2014) [35] | USA/Health centre for LGBT communities | 3 (60 mins) tailored personal counselling sessions to enhance risk reduction vs standard of care: range of standard of care support services for newly-diagnosed | 102 newly diagnosed HIV positive GBMSM (51 intervention, 51 control) | June 2009— May 2011 | 3, 6 and 9 months, 1 year in total | # condomless AI occasions (insertive or receptive); condomless AI with serodiscordant partners; # unprotected condomless AI with serodiscordant partners (past 3 months) | 1)# condomless AI acts |
| colspan 8: **Group interventions for GBMSM (HIV negative, untested or regardless of HIV status)** |
| Harawa et al (2013) [36] | USA/Not stated | Intervention (MAALES): 6 x 2-hour small-group intervention over 3 weeks, with booster sessions at 6 & 18 weeks vs Control: standard client-centred HIV education & risk-reduction session (15–25 mins) | 386 black bisexual men (198 intervention, 188 control) | Aug 2007 – May 2011 | 3 and 6 months | # male, female, trans partners; # episodes of any anal or vaginal intercourse, any unprotected intercourse (and serodiscordant) (past 90 days) | 1)# condomless AI acts 2)# partners |
| Hidalgo et al (2015) [37] | USA/LGBT community health centre | Intervention: 6 sessions of MyPEEPS (Male Youth Pursuing Empowerment, Education and Prevention around Sexuality), group-level (5–10) intervention to reduce sexual risk vs time-matched group-level didactic control | 101 young HIV negative GBMSM (58 intervention, 43 control) | 18 months to Dec 2010 | 6 and 12 weeks | Male-male sexual risk: # sex partners # condomless AI sex partners, frequencies condomless AI or oral sex, sex (oral or anal) or condomless AI under the influence of alcohol/drugs). (past 6 weeks) | 1)Any condomless AI (yes/no) 2)# partners |
| Kurtz et al (2013) [38] | USA/Field offices | 4-session small group (5–10) sexual & SU risk reduction intervention vs single session individual control including risk assessment & risk reduction counselling | 515 substance-using GBMSM (252 intervention, 263 control) | Recruitment: Nov 2008— Oct 2010 | 3, 6 and 12 months | Frequency of condomless AI involving HIV transmission risk (excluding condomless AI if both HIV positive) (past 90 days) | 1)# condomless AI acts 2)# partners |
| O'Donnell et al (2014) [39] | USA/Research sites | Sin buscar excusas (SBE): 45-60-min single-session intervention (in language of choice) for Latino GBMSM (4–9 per group) vs Control: non-attention condition (no group activity) | 370 enrolled (190 intervention, 180 control) | Enrolment: Aug 2008 – Aug 2009 | 3 months | # condomless AI acts with last 2 male partners, condom use at last intercourse with male (0/1), self-report of HIV test during (past 3 months) | 1)Condom use at last sexual intercourse (yes/ no) |
| Rhodes et al (2017) [40] | USA/ Community organisation & business meeting space | HOLA en Grupos: 4-session (4 hrs each) Spanish-language small-group intervention vs Control: attention equivalent health education with same # sessions | 304 Hispanic / Latino GBMSM (152 intervention, 152 control) | Enrolment Dec 2012 – Feb 2015 | 6 months | Consistent condom use (at every insertive or receptive AI with men & insertive vaginal or anal with women (past 3 months) | 1)Consistent condom use (every instance of AI with a man = yes/ not every instance of AI with a man = no) |
| colspan 8: **Group interventions for GBMSM (living with diagnosed HIV)** |
| Williams et al (2013) [41] | USA/Not stated | ES-HIM: stress-focused sexual risk reduction intervention vs HP: general health promotion intervention—6 small-group (2-hr) sessions, over 3 weeks by ethnically matched male facilitator | 117 HIV positive bisexual black men with childhood sexual abuse history | Study duration: 2007–2011 | 3 and 6 months | Sexual risk behaviours: #unprotected receptive or insertive anal intercourse acts, #partners, psychological symptoms, stress biomarkers (past 3 months) | 1)# condomless AI (receptive) events 2)# partners |

*(Continued)*

**Table 1.** (Continued)

| Study (year) (ref) | Country/ Setting | Intervention | Participants | Recruitment | Follow -up | Primary outcome(s) (recall period) | Chosen outcome(s) (at end of follow-up period) for data extraction |
|---|---|---|---|---|---|---|---|
| Hart et al (2021) [53] | Canada/ Community setting and online | GPS:, a community-based and peer-delivered sexual health promotion group intervention for HIV + GBM: weekly 2-hour group sessions over 8 weeks, led by two HIV + gay, peer facilitators | 183 HIV diagnosed MSM | Not stated | 3 and 6 months | prevalence of CAS with HIV-negative and unknown HIV status partners (past 2 months) | 1) CLAI with serodiscordant partners (yes/no) |
| **Online interventions for GBMSM (HIV negative, untested or regardless of HIV status)** | | | | | | | |
| Christensen et al (2013) [42] | USA/Online | SOLVE: 2-level virtual world simulating common obstacles to safer sex– exposed to intervention if makes risky choice vs waitlist control | 921 HIV negative GBMSM (437 intervention, 484 control) | Enrolment: Feb–Nov 2012 | 3 months | Change in counts of risky sexual behaviour (including #condomless AI events) (past 3 months) | 1)# condomless AI events |
| Fernandez et al (2016) [43] | USA/Research offices & online | POWER & HEALTH online via live chat by facilitators; POWER: 3 individual sessions (60–90 mins, weekly for 3 weeks) with facilitator on HIV risk & protection, motivation & skills vs HEALTH (control): 1 x 3–4 hr session on health issues for black men | 211 bisexual black men (108 intervention, 103 control) | Enrolment: June 2011— Nov 2012 | 3 months | Condomless vaginal or anal intercourse (past 3 months) | 1)# condomless AI events |
| Lau et al (2016) [54] | Hong Kong/ Online | SC: STD-related cognitive approach (2 x 5 min videos), SCFI: STD-related cognitive plus fear approach (1 x 10 min video), Control: HIV-related information-based approach | 402 GBMSM (133 SC, 133 SCFI, 136 control) | Not stated | 3 months | condomless AI with any male partner, with regular male partner, with casual partner, with commercial sex partner (past month) | 1)Condomless AI (any male sex partner) (yes/ no) |
| Mustanski et al (2013) [44] | USA/Online | Keep it up (KIU!) intervention (7 modules across 3 sessions, ~2 hours in total) vs online didactic HIV knowledge control with same # modules & sessions | 102 young HIV negative GBMSM (50 intervention, 52 control) | Aug 2009— Sep 2010 | 6 and 12 weeks | # condomless AI acts (past 6 weeks) | 1)# condomless AI acts |
| Cruess et al (2018) [50] | USA/Online | HIV Internet Sex study (HINTS) 4 online group sessions, 45 min in duration, occurred in sequential order (Sessions 1–4) over 2 weeks using IMB for sexual risk reduction Vs time matched control of healthy living info | 167 HIV diagnosed GBMSM (85 intervention, 82 control) | Not stated | 6 months | # condomless AI acts (past 6 months) | 1)# condomless AI acts 2)# partners |
| Hightow-Weidman et al (2019) [23] | USA/Online | HealthMpowerment.org (HMP) -a mobile optimized, online intervention Vs information-only control website | 474 young Black GBMSM (HIV diagnosed and undiagnosed) | Nov 2013 and October 2015 | 3 and 6 and 12 months | Condomless AI (past 3 months) | * |
| Chiou et al (2020) [55] | Taiwan/Online | Use of the Safe Behaviour and Screening (SBS) app for 6 months Vs no app | 265 HIV negative GBMSM | August 2015 – May 2017 | 6 months | AI and condom use during AI (past 3 months) | 1)Mean percentage of condom use during AI |

*(Continued)*

**Table 1.** (Continued)

| Study (year) (ref) | Country/ Setting | Intervention | Participants | Recruitment | Follow -up | Primary outcome(s) (recall period) | Chosen outcome(s) (at end of follow-up period) for data extraction |
|---|---|---|---|---|---|---|---|
| *Online interventions for GBMSM living with HIV* | | | | | | | |
| Milam et al (2016) [45] | USA/Online | Monthly (x 12) brief, computer accessed, sexual behaviour survey (control), vs survey plus Internet-delivered tailored messages about safer-sex, disclosure & initiation of ART (intervention) | 179 HIV positive GBMSM (90 intervention, 89 control) | Nov 2010—July 2012 | Monthly for 12 months | Cumulative STI incidence (serological testing) (past 12 months) | 1)STI incidence |
| Hirshfield et al (2019) [51] | USA/Online | Sex Positive! is a two-arm, video-based web intervention. Men in each arm received 6 weekly videos after completing a baseline assessment and 4weekly booster videos following a 6-month assessment | 830 HIV positive MSM(413 intervention, 417 control) | 2015 | 12 months | reduction in the number of serodiscordant CAS partners (past 3 months) | 1)condomless AI change (with unknown serodiscordant partners) (yes/no) |
| *Contingency management (CM) for substance use* | | | | | | | |
| Landovitz et al (2014) [46] | USA/Not stated | 8-week behavioural interventions–CM: voucher-based incentives for thrice-weekly stimulant-free urine samples vs Noncontingent "yoked" control (NCYC): incentives not tied to substance abstinence | 140 stimulant-using HIV negative GBMSM (70 intervention, 70 control) | Enrolment: June 2010—June 2012 | 3 and 6 months | PEP initiators on time from exposure to first dose, medication adherence, course completion, condomless AI (past 6 months) | 1)condomless AI events |
| Nyamathi et al (2017) [47] | USA/ Community sites | Nurse case management (NCM) + CM vs Standard education (SE) + CM (control). NCM: 16-week program of 8 x 20-min one-to-one & 8 group (4–5) or individual peer sessions; SE: once (20-min session) by health educator; CM: urine samples 3x / week for 16 weeks | 422 homeless, stimulant-using GBMSM (221 intervention, 211 control) | Not stated | 4 and 8 months | Stimulant use & multiple partners (two or more) (past 30 days) | 1)multiple partners (2 or more partners = yes/ 1 or no partners = no) |
| *HIV Pre-exposure Prophylaxis (PrEP)* | | | | | | | |
| Mayer et al (2017) [48] | USA/Primary care clinics | CBT: 4 weekly & 2 booster 50-min sessions (2 & 3 months after PrEP initiation) of "Life-Steps for PrEP", nurse-delivered, CBT-oriented PrEP adherence vs Information and support counselling (ISP): time & session matched counselling control | 50 HIV negative GBMSM (intervention 25, control 25) | Nov 2012—June 2014 | 3 and 6 months | PrEP adherence (via electronic real-time adherence monitoring), quantification of plasma tenofovir levels, condomless AI (recall period not stated) | 1)consistent condom use (<100%condom use = yes/100% condom use = no) |
| McCormack et al (2016) (PROUD) [52] | UK/13 Sexual health clinics | Daily combined TDF-FTC (245 mg TDF, 200 mg FTC) immediately vs after 1-year deferral | 544 HIV negative GBMSM (275 intervention,269 deferred control) | Enrolment: Nov 2012 – Apr 2014 | Quarterly over 2 years | HIV diagnosis Condomless AI Number of partners (past 3 months) | 1)HIV incidence 2)STI incidence 3)(receptive) condomless AI with > = 10 partners (yes/no) 4)# partners |

*(Continued)*

**Table 1.** (Continued)

| Study (year) (ref) | Country/ Setting | Intervention | Participants | Recruitment | Follow -up | Primary outcome(s) (recall period) | Chosen outcome(s) (at end of follow-up period) for data extraction |
|---|---|---|---|---|---|---|---|
| Molina et al (2016) (ANRS IPERGAY) [57] | France & Canada/Clinics (6 in France, 1 in Canada) | Combined TDF-FTC: fixed-dose (300 mg TDF, 200 mg FTC); vs placebo. Two pills with food 2–24 hours before sex, followed by third pill 24 hours after the first drug intake and a fourth pill 24 hours later. | 400 HIV negative GBMSM (199 intervention, 201 placebo control) | Enrolled: Feb 2012 –Oct 2014 | 4 and 8 weeks, then every 8 weeks for 24 months. | HIV diagnosis | 1)HIV incidence |
| Liu et al (2013) [49] | USA/Clinics across San Francisco, Atlanta, and Boston | Tenofovir disoproxil fumarate or placebo at enrollment or after a 9-month delay and followed for 24 months | 400 HIV-negative GBMSM | February 2005 —July 2007 | 6 and 12 and 18 and 24 months | # of sex partners and UAI (past 3 months) | 1)condomless AI events |
| Chaix et al (2018) (ANRS IPERGAY) [59] | France & Canada/Clinics (6 in France, 1 in Canada) | Combined TDF-FTC: fixed-dose (300 mg TDF, 200 mg FTC); vs placebo. Two pills with food 2–24 hours before sex, followed by third pill 24 hours after the first drug intake and a fourth pill 24 hours later. | 400 HIV negative GBMSM (199 intervention, 201 placebo control) | Enrolled: Feb 2012 –Oct 2014 | 4 and 8 weeks, then every 8 weeks for 24 months. | Herpes simplex virus (HSV)-1/2 incidence | 1)STI incidence |
| Molina et al (2018) (ANRS IPERGAY) [58] | France & Canada/Clinics (6 in France, 1 in Canada) | Combined TDF-FTC: fixed-dose (300 mg TDF, 200 mg FTC); vs placebo. Two pills with food 2–24 hours before sex, followed by third pill 24 hours after the first drug intake and a fourth pill 24 hours later. | 232 HIV negative GBMSM | July 2015— Jan 2016, | Every 2 months until June 2016 | New STI | 1)STI incidence |

*unable to extract data for any chosen outcome

**unable to extract data on primary outcome

ref:reference

AI: Anal intercourse

OR: odds ratio, aOR: adjusted odds ratio RR: relative risk

CBT: cognitive behavioural therapy

CM: contingency management

CDC: United States Centers for Disease Control and Protection

MI: Motivational Interviewing

GBMSM: gay and bisexual men who have sex with men

MSMW: men who have sex with men and women

PCC: Personalized Cognitive Counselling

PEP: post exposure prophylaxis PrEP: pre-exposure prophylaxis

SC: STD-related cognitive approach

SCFI: STD-related cognitive plus fear appeal imagery approach

STI: Sexually Transmitted Infection

TDF-FTC: Combined dose tenofovir disoproxil fumarate (TDF) and emtricitabine (FTC)

USA: United States of America UK: United Kingdom

3 MV: Many Men, Many Voices

management. Fig 2 presents the results of the two PrEP studies [52] that evaluated the efficacy of PrEP on HIV incidence and both demonstrated a strong and significant reduction [52, 57]

**Table 2. Results, quality score and calculated standardised effect sizes for chosen outcomes of randomised controlled trials included in this review.**

| Study (year) | Overall Results as reported in publication (abstract or results) | Quality Score** | Calculated Standardised Effect Sizes for chosen outcomes* |
|---|---|---|---|
| Coffin et al (2014) | Intervention vs control: No significant between-group differences were found in the three primary study outcomes: number of unprotected anal intercourse events (UAI), number of UAI partners, and UAI with three most recent non-primary partners. | + | Not Estimable |
| Eaton et al (2017) | Intervention vs control: proportion of condom-protected sex acts significant higher (p<0.05), condomless insertive sex significantly lower (p>0.01), condomless receptive sex (p>0.05); No overall effect on urine STI diagnoses. | ++ | 1)# condomless AI acts: Cohen's *d*:-1.49 95%CI: -1.57, -1.41<br><br>2)# partners: Cohen's *d* -0.32 95%CI: -0.40, -0.24 |
| Jemmott et al (2015) | Although the intervention did not affect the proportion of condom-protected intercourse acts, unprotected sexual intercourse, multiple partners, or insertive anal intercourse, it did reduce receptive anal intercourse compared with the control. | ++ | 1)Consistent condom use: SES: 0.01 95%CI: -0.19, 0.20<br>2)# partners SES: -0.02 95%CI: -0.19, 0.14 |
| Lauby et al (2016) | Participants in the intervention group were more likely than control participants to report a small/ moderate decrease (p = 0.014) or a large decrease (p = 0.017) in episodes without condoms with male and female partners combined. No effect for # male partners (p = 0.101) | + | 1)# condomless AI acts *0.17 95%CI: 0.00, 0.34<br>2)# partners *SES: -0.02 95%CI: -0.19, 0.14 |
| Pachankis et al (2015) | Intervention vs waitlist: significantly reduced past-90-day condomless sex with casual partners (p<0.001) | + | Not Estimable |
| Parsons et al (2014) | Regardless of condition, participants reported significant reductions in UAI and substance use over time. Intervention group less likely to report condomless AI vs education group (p = 0.0001) | + | 1) # condomless AI events: SES: -0.15 95%CI: -0.21, -0.09 |
| Crosby et al (2018) | Significant intervention effects relative to incident sexually transmitted diseases were not observed. However, HIV diagnosed MSM reported greater odds of consistent condom (p = 0.001), and HIV negative MSM reported twice the odds of consistent condom use (p< 0.001), compared to control, in receptive anal sex over 12 months. | ++ | 1)STI incidence SES: 0.09 95%CI: -0.03, 0.21<br>2) Consistent condom use (yes/no) SES:0.42, 95%CI 0.31, 0.53 |
| Llewellyn et al (2019) | There were no significant impacts on sexual risk behaviour or any of the psychological measures, and no discernible reduction in requests for repeat PEP or rates of STIs within a year. | + | Not Estimable |
| Mimiaga et al (2019) | At the 6-month post-intervention visit intervention participants reported 1.1 CAS acts with men who were HIV-infected or whose status they did not know compared to 2.8 among control participants (p< 0.0001) at 6 months. | + | 1)# condomless AI events (with HIV+ or unknown status partner) *SES: -1.21 95%CI: -1.03, -0.33 |
| O'Cleirigh et al (2019) | At the follow-up visits, treatment condition had significant reductions in the odds of any CAS and reductions in CAS. Treatment condition experienced a significantly steeper decrease in sexual risk over time compared to those in the control condition (p = .04). | + | 1)# condomless AI events (with HIV+ or unknown status partner) *SES: -0.47 95%CI: -0.77, -0.17 |
| Reback et al (2019) | Only participants in TXT-PHE and TEXT-Auto arms reduced CAI with main male partners, and only TEXT-Auto participants reduced CAI with anonymous male partners | + | 1)# condomless AI events (with casual partner) *SES: 0.28 95%CI: 0.13, 0.43 |
| Wray et al (2019) | Intervention Vs control: participants reported fewer high-risk condomless anal sex events than controls, but these differences were not significant. | + | Not Estimable. |
| Nöstlinger et al (2016) | Intervention vs control reported lower transmission risk at 3 months (9.01, CI: 1.78 to 45.71; P = 0.008), but not significant at 6 months (1.31, CI: 0.38 to 4.54; P = 0.67) | - | 1)Condom use at last intercourse (yes/no) SES:0.14 95%CI: -0.32, 1.03 |
| Schwarcz et al (2013) | The mean number of UAI episodes declined in both groups at 6 months, declined further in the PCC group at 12 months, while increasing to baseline levels among controls; these differences were not statistically significant. No differences were observed in STI incidence between the two groups. | + | 1)STI incidence (new gonorrhoea) SES:-0.26 95%CI: -1.05, 0.55 (new chlamydia) SES: -0.6 95%CI: -0.06, 0.14 |
| Sikkema et al (2014) | Intervention participants significantly reduced the frequency of UAI with HIV serodiscordant (HIV negative or status unknown) partners over the 9-month follow-up period significant. | + | 1)# condomless AI acts *SES: -1.22 95%CI: -1.41, -1.03 |
| Harawa et al (2013) | Adjusted results indicated significant intervention-associated reductions in the numbers of total anal or vaginal sex acts. Near significant reductions were observed for number of male intercourse partners. | + | 1)# condomless AI acts *SES: -0.03 95%CI: -0.15, 0.08<br>2)# partners *SES: -0.12 95%CI: -0.23, 0.00 |

*(Continued)*

**Table 2.** (*Continued*)

| Study (year) | Overall Results as reported in publication (abstract or results) | Quality Score** | Calculated Standardised Effect Sizes for chosen outcomes* |
|---|---|---|---|
| Hidalgo et al (2015) | Over the entire follow-up period, intervention participants were less likely than controls to engage in any sexual behaviour while under the influence of substances (p<0.05), and also observed in this group was a decreasing trend of unprotected anal sex while under the influence of substances (p = .08). Follow-up differences between groups on social cognitive outcomes favoured the intervention group, though these differences were non-significant. | - | 1)Condomless AI (yes/no) SES: 0.06 95%CI -0.48, 0.59 2)# partners SES: -0.02 95%CI (-0.36, 0.32) |
| Kurtz et al (2013) | Effect sizes for sexual risk and substance use outcomes were moderate to large. No significant differences in outcome between the experimental and control conditions were observed. | + | 1)# condomless AI acts *SES: -0.11 95%CI: -0.2, -0.02 2)# partners *SES: 0.04 95%CI: -0.05,0.13 |
| O'Donnell et al (2014) | At a three-month follow-up, there was a sharper decrease in unprotected intercourse in the intervention group compared to controls (p<0.05) | + | 1) Condom use at last sex: SES: 0.29 95%CI: 0.01, 0.57 |
| | Intervention participants also reported more condom use at last intercourse (p<0.02). | | |
| Rhodes et al (2017) | Intervention participants reported increased consistent condom use during the past 3 months (p < .001) | ++ | 1)Consistent condom use (yes/no) SES: 0.77 95%CI: 0.37, 1.15 |
| Williams et al (2013) | Both interventions decreased and sustained reductions in sexual risk and psychological symptoms. The stress-focused intervention was more efficacious than the general health promotion intervention in decreasing unprotected anal insertive sex and reducing depression symptoms. | + | 1)# condomless AI (receptive) events *SES: -0.02 95%CI: -0.23, 0.19 2)# partners *SES: 0.06 95%CI: -0.15, 0.26 |
| Hart et al (2021) | GPS prevention counseling demonstrated a 43% relative reduction at 3-month follow-up in CAI with serodiscordant partners and significant reductions in sexual compulsivity | ++ | 1) CLAI with serodiscordant partners (yes/no) SES:-0.21 95%CI: -0.53, 0.11 |
| Christensen et al (2013) | Direct effect of intervention on condomless AI not significant. Intervention group reported greater reductions in shame, which predicted reductions in risky sexual behaviour at follow-up. | + | 1)# condomless AI events *SES: -0.07 95%CI: -0.13, 0.00 |
| Fernandez et al (2016) | The intervention was associated with significantly lower odds of condomless anal intercourse with male partners (p = 0.020) but not with female partners and serodiscordant sex with male partners but not with female partners | + | 1)# condomless AI events SES:-0.33 95%CI: -0.6, -0.05 |
| Lau et al (2016) | No statistically significant differences across the three groups for condomless AI at month 3. | + | 1)Condomless AI (yes/no) SES: -0.03 95%CI: -0.3, 0.24 |
| Mustanski et al (2013) | Compared to the control condition, participants in the intervention arm had a 44% lower rate of unprotected anal sex acts at the 12-week follow-up (p < 0.05). | ++ | 1)Condomless AI events *SES: -0.27 95%CI:-0.46, -0.07 |
| Cruess et al (2018) | HINTS intervention did not have a significant impact on frequency of CAS when examining sexual risk behaviour across all male partners, however there were significant intervention effects when tested separately by partner serostatus. | ++ | 1)# condomless AI events SES: -0.15 95%CI:-0.3, 0.001 2)# partners SES: 0.25 95%CI: 0.09, 0.4 |
| Hightow-Weidman et al (2019) | The rate of self-reported condomless anal intercourse (CAI) at 3-months was 32% lower in the intervention group compared to the control group, however this effect was not sustained at 12 months. | + | Not Estimable |
| Chiou et al (2020) | Compared to the control group, the experimental group had significantly higher mean score of safe behaviour knowledge, motivation, and skills; percentage of condom use during anal intercourse; frequency of searching for testing resources and getting HIV and syphilis tests. | - | 1)Consistent condom use (yes/no) *SES: 5.56 95%CI: 5.44, 5.68 |
| Milam et al (2016) | In a modified intent to treat analysis, there was no difference in 12-month STI incidence between the intervention and control arms (30 vs. 25%, respectively; p = 0.5). | ++ | 1)STI incidence SES: 0.17 95%CI: -0.21, 0.55 |
| Hirshfield et al (2019) | At 3-month follow-up, men in the intervention arm reported significantly reduced risk of having unknown serodiscordant CAI partners than men in the control arm (RR 0.60, 95% CI 0.39–0.92), partially supporting study hypotheses. Aside from this finding, similar reductions in sexual risk behaviors were observed in both arms over the study period | + | 1) condomless AI change (with unknown serodiscordant partners) (yes/no) SES: -0.08 95% CI: -0.34, 0.18 |

(*Continued*)

**Table 2.** (Continued)

| Study (year) | Overall Results as reported in publication (abstract or results) | Quality Score** | Calculated Standardised Effect Sizes for chosen outcomes* |
|---|---|---|---|
| Landovitz et al (2014) | PEP course completion was greater in the CM group vs the NCYC group, with a trend towards improved medication adherence in the CM group. No significant difference in condomless AI at 6 months. | + | 1)condomless AI events *SES: 0.15 95%CI: -0.03, 0.34 |
| Nyamathi et al (2017) | No significant group or group-by-time effects in number or percentage of positive urine drug screens at each point in study. Rates of decline in multiple partners were not significant. | - | 1)multiple partners (yes/no) SES: -0.14 95%CI: -0.48, 0.19 |
| Mayer et al (2017) | No significant differences in odds of adherence (OR 0.7, [0.3–1.7], p = 0.48); plasma tenofovir levels significant higher in CBT vs ISP at 6 months using imputation (p = 0.037), no significant differences in condomless AI (p = 0.47) | + | 1)consistent condom use (yes/no) SES:-0.06 95%CI: -0.33, 0.21 |
| McCormack et al (2016) (PROUD) | Three HIV infections occurred in the immediate group (1·2/100 person-years) versus 20 in the deferred group (9·0/100 person-years) despite 174 prescriptions of post-exposure prophylaxis in the deferred group (relative reduction 86%, p = 0·0001; absolute difference 7·8/100 person-years, 90% CI 4·3–11·3). | ++ | 1)HIV incidence SES:-1.06 95%CI: -2.02,-0.41 2)STI incidence SES:0.04 95%CI: -0.14, 0.21 3)(receptive) condomless AI (yes/no) SES:-0.06 95%CI: -0.33, 0.21 4)# partners SES:0.11 95%CI: -0.1.0.33 |
| Molina et al (2016) (ANRS IPERGAY) | A total of 16 HIV-1 infections occurred during follow-up, 2 in the TDF-FTC group (incidence, 0.91 per 100 person-years) and 14 in the placebo group (incidence, 6.60 per 100 person-years), a relative reduction in the TDF-FTC group of 86% (p = 0.002). | ++ | 1)HIV incidence SES:-1.07 95%CI: -2.33, -0.28 |
| Liu et al (2013) | Mean numbers of partners and proportion reporting unprotected anal sex (UAS) declined during follow- up (p<0.05), and mean UAS episodes remained stable. | ++ | 1)condomless AI events SES:-0.03 95%CI: -0.21,0.14 |
| Chaix et al (2018) (ANRS IPERGAY) | Overall HSV-1 incidence was 11.7 per 100 person-years; 16.2 and 7.8 per 100 person-years in the TDF/FTC and placebo arm, respectively (P = 0.19). Overall HSV-2 incidence was 7.6 per 100 person-years; 8.1 and 7.0 per 100 person-years in the TDF/ FTC and placebo arm, respectively (P = 0.75). On-demand oral PrEP with TDF/FTC failed to reduce HSV-1/2 incidence in this population. | ++ | 1)STI incidence HSV1 SES:0.38 95%CI: -0.26, 1.04 HSV2 SES:0.08 95%CI:-0.47, 0.66 |
| Molina et al (2018) ANRS IPERGAY | The occurrence of a first STI in participants taking PEP was lower than in those not taking PEP (p = 0·008). Similar results were observed for the occurrence of a first episode of chlamydia (p = 0·006) and of syphilis (p = 0·047); for a first episode of gonorrhoea the results did not differ significantly (p = 0·52). | ++ | 1)STI Incidence SES -0.38 95%CI:-0.69, -0.07 |

*A negative effect size indicates that the outcome was less frequent in the intervention group compared to the control group.

** ++ = high quality + = mixed quality - = poor quality

AI: Anal intercourse UAI: Unprotected anal intercourse CAS: condomless anal sex, CLAI: condomless anal intercourse, UAI:Unprotected anal intercourse, UAS: Unprotected anal sex, CAI: Condomless anal intercourse

OR: odds ratio aOR: adjusted odds ratio sOR: standardised odds ratio, CI: Confidence interval, RR: relative risk *Cohens *d*

CM: contingency management CBT: cognitive behavioural therapy ITT: Intention to treat

MSM: men who have sex with men MSMW: men who have sex with men and women

PEP: post exposure prophylaxis, PrEP: pre-exposure prophylaxis TDF-FTC: Combined dose tenofovir disoproxil fumerate (TDF) and emtricitabine (FTC)

STI: Sexually transmitted infection HSV: Herpes Simplex Virus

HINTS: HIV Internet Sex study SC: STD-related cognitive approach, SCFI: STD-related cognitive plus fear appeal imagery approach

compared to no or delayed PrEP. The other four papers in which PrEP was the intervention reported outcomes of CLAI [48, 49], or STI incidence [58, 59].

Fig 3 presents the results of the six papers (one paper reported separate results for HSV1 and HSV2 [59], and another paper reported separate results for chlamydia and gonorrhoea [34]) that looked at the intervention effects on STI incidence. Only one intervention that included

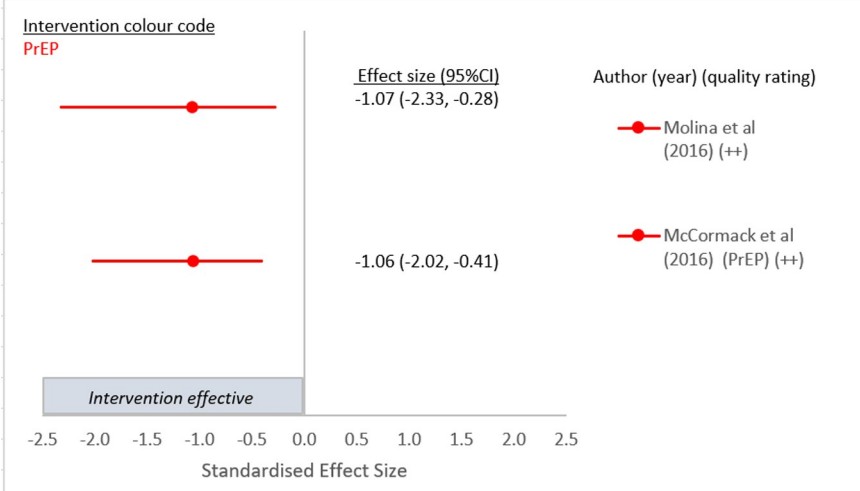

**Fig 2. Intervention effect on HIV incidence.**

PrEP plus a single oral dose of 200 mg doxycycline post exposure prophylaxis (PEP) within 24 hours after sex compared to no prophylactic dose of doxycycline showed a significant reduction in STI incidence in the intervention arm compared to no intervention [58] (Fig 3).

Fig 4 demonstrates the results for the twenty-eight papers that examined the different intervention effects on sexual behaviour (measures of condom use or CLAI, and/or partner numbers). Two studies of group interventions among Latino or Hispanic GBMSM in the USA [39, 40], one online intervention [55] and a one-to-one counselling intervention [30] demonstrated a significant increase in consistent condom use, and five studies that used one-to-one counselling as an intervention demonstrated significant reductions in CLAI [26, 29, 31, 32, 35] (Fig 4). However two studies that used one-to-one counselling as an intervention [28, 33] and one that used PrEP [52] as an intervention actually demonstrated a small but significant increase in measures of CLAI after standardised effect sizes were calculated [28] (Fig 4).

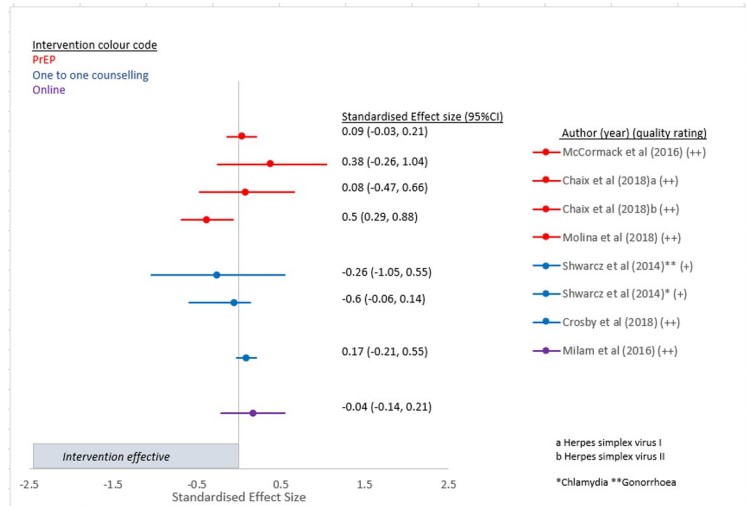

**Fig 3. Intervention effect on STI incidence.**

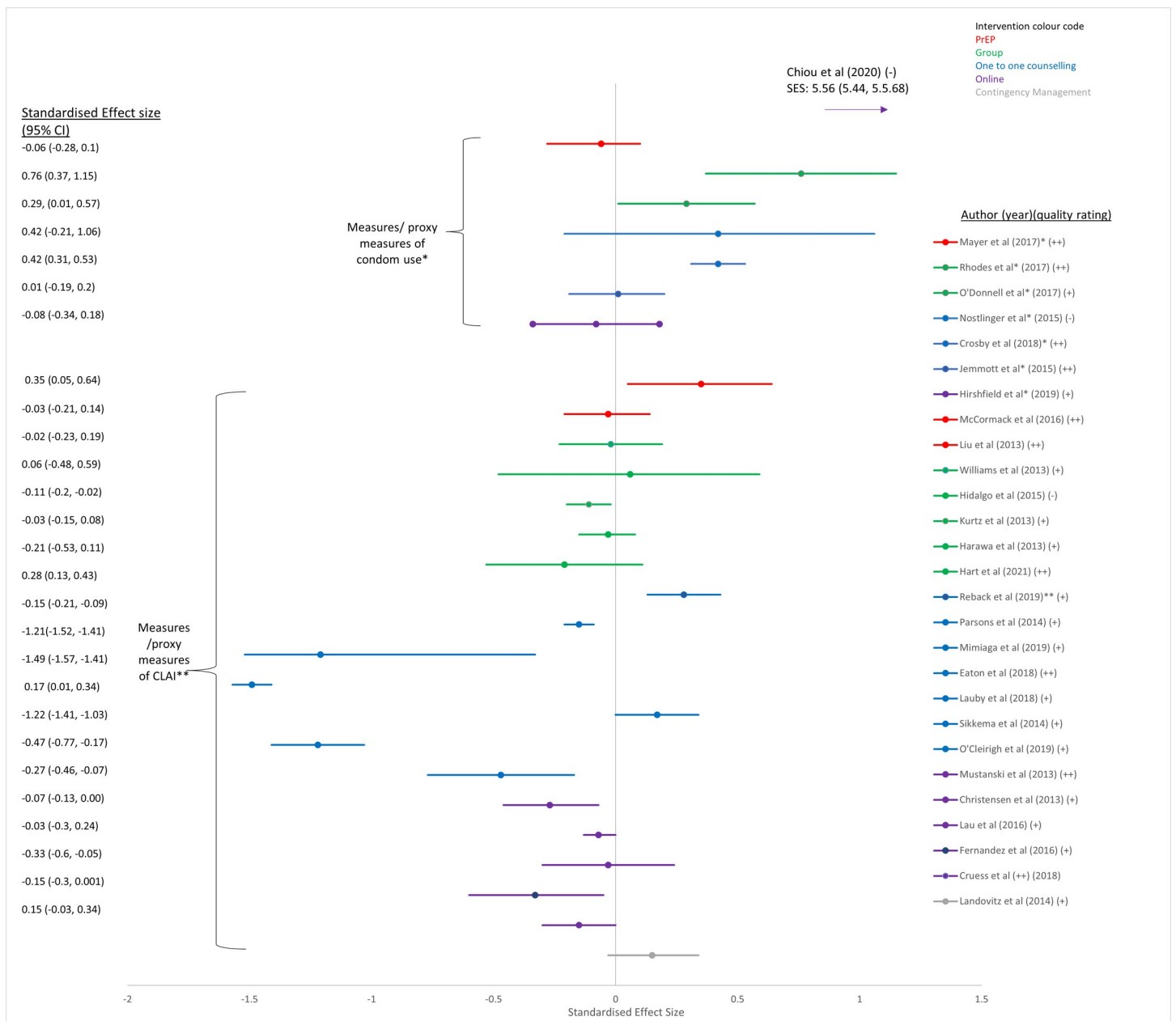

**Fig 4. Intervention effect on measures of CLAI or condom use.** *outcome reported as a measure of condom use–positive values reflect a desirable outcome ** outcome reported as a measure of CLAI–negative values reflect a desirable outcome.

Fig 5 presents the results of the ten papers that examined the different intervention effects on partner numbers. Only one study that used one-to-one counselling as an intervention demonstrated a significant reduction in partner numbers [26] whilst the other interventions did not reduce partner numbers significantly [27, 28, 36–38, 41, 47, 52], and one online intervention actually demonstrated a significant increase in partner numbers [50] (Fig 5).

## Intervention assessment

**One-to-one counselling.** Fifteen studies evaluated the efficacy of one-to-one counselling interventions (Table 1 (colour blue in Figs 2–5)). Twelve studies included GBMSM with

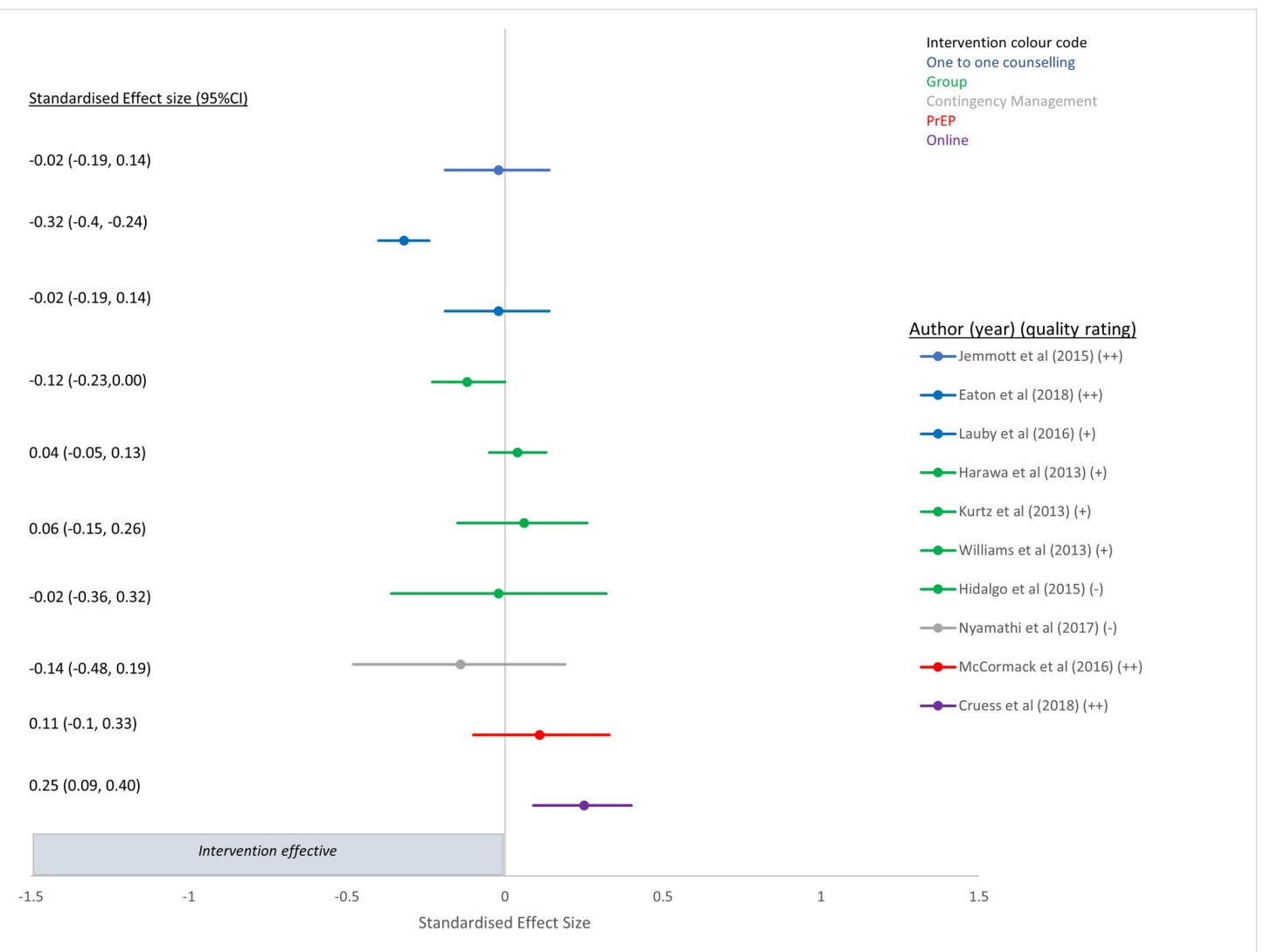

**Fig 5. Intervention effect on partner numbers.**

negative or unknown HIV status, or regardless of HIV status [20–22, 24, 26–33] and three were aimed at HIV positive men [34, 35, 56]. Interventions ranged from providing a single, counsellor-delivered, 45-min session to HIV negative men highlighting the limitations of sero-sorting [26], to ten sessions of cognitive behavioural therapy for trauma and self-care delivered by clinical psychologists and pre- and post-doctoral fellows in clinical psychology [32]. The majority of the studies were conducted in the USA [20, 21, 24, 26–35], one in the UK [22] and one had multiple sites across Europe [56]. Only four studies were rated as high quality [22, 26, 27, 30] (Table 2). There was inconsistent evidence about the efficacy of one-to-one counselling as a behavioural intervention. In terms of outcomes, none of them reported on HIV incidence, two on STI incidence [30, 34], ten on measures of condom use or CLAI [26–33, 35, 56] and three on partner numbers [26–28]. Of the two studies that reported on STI incidence, one was rated as high quality [30] and the other mixed quality [34], but neither found that the intervention had a statistically significant effect on the outcome. Of the ten papers that investigated the effect of one to one counselling on measures of condom use of CLAI, six demonstrated

statistically significant reductions in CLAI [26, 29, 30, 31, 32, 35], of which two were judged to be high quality by reviewers [26, 30]. One study used a single-session risk decision intervention to highlight misbeliefs about sexual behaviour among HIV negative GBMSM and had a large overall standardised effect size of -1.49 (95%CI -1.57, -1.41) (mean number of condomless sex acts(last 3 months) in intervention group 1.9 at 12 months vs 3.1 in control group) [26] (Fig 4). The other study used a single session, tailored, interactive programme to educate and promote condom use in young, black GBMSM, and reported a moderate standardised effect size of 0.42 (95%CI 0.31, 0.53) (likelihood of consistent condom use for receptive anal sex among HIV negative individuals) [30] (Fig 4). Although there were two other RCTs of a one-to-one counselling intervention that were assessed as high quality [22, 27], neither demonstrated a statistically significant result and one provided limited results. The first used a targeted HIV/STI risk reduction intervention (targeting condom use) over three sessions and examined consistent condom use among African American GBMSM (regardless of HIV status) [27] and found no significant difference between experimental and control arm at 12 month follow up for a reduction in either CLAI (SES: 0.01 95%CI-0.19, 0.20) (Fig 4) or partner numbers (SES -0.02 95%CI -0.19,0.14) (Fig 5). The second study used two telephone sessions of augmented motivational interviewing (MI) to reduce risky sexual behaviour in GBMSM prescribed PEP. The results reported in the paper demonstrated no significant impacts on sexual risk behaviour or any of the psychological measures, and no discernible reduction in requests for repeat PEP or rates of STIs within a year [22]. Lack of information provided in this latter paper meant that we could not produce a SES (and so this is not presented in the Figure). Of the three studies that reported on number of partners, two were rated as high quality [26, 27] whilst the other was rated as mixed quality [28]. One of the high quality studies demonstrated that a single one to one counselling session (45 minutes) that highlighted misbeliefs using graphic novel and sexual network diagrams, demonstrated a small but significant effect on partner numbers at 12 months (from 1.75 (past 3 months) in the control arm to 1.65 in the intervention arm) (SES -0.32 95%CI:-0.4,-0.24), despite not having any overall effect on STI outcomes [26].

**Group interventions.** Seven group intervention trials were included in this review: six were conducted in the USA (Table 1 (colour green in Figs 2–5)) [36–41] and one in Canada [53]. Five studies were aimed at HIV negative or undiagnosed GBMSM and two were aimed at HIV positive men [41, 53]. The interventions delivered theory-driven, interactive, behaviour change group sessions (ranging from one to eight sessions) in-person designed to reduce CLAI or increase condom use. The majority of the studies recruited sub-groups of GBMSM; two recruited black bisexual men [36, 41], two were targeted at Latino GBMSM [39, 40], one intervention was aimed at young HIV negative GBMSM [37], and one at substance using GBMSM [38]. None of them reported on HIV incidence or STI incidence, all seven reported on measures of CLAI or condom use [36–41, 53] and three found a statistically significant and beneficial effect on condom use [39, 40] or reduction in CLAI [38]. Four reported on partner numbers [36–38, 41], however only one found the intervention to be borderline effective at reducing partner numbers (SES -0.12, 95%CI: -0.23, 0.00) (from 2.49 partners at baseline in the intervention group to 1.04 at 6 month follow up, and 1.91 partners at baseline in the control group to 1.50 partner at 6 month follow-up) [36]. Two of these RCTs were rated as high quality [40, 53], four were of mixed quality [36, 38, 39, 41] and one was rated as poor quality [37] (Table 1). The first group study that demonstrated a significant increase in condom use used a four-session, 16-hour, Spanish language group intervention that showed four-fold difference in odds of consistent condom use between intervention and control [40] (SES 0.76 95%CI 0.37, 1.15) at 6 months follow up and was rated as high quality (Fig 4). Another RCT, rated mixed quality, was aimed at Latino GBMSM and found that a single-session group intervention increased condom use at last sex compared with non-group activity control at 3

months follow-up (SES 0.29 95%CI 0.01, 0.57) (Fig 4) [39]. The remaining interventions did not demonstrate any significant efficacy for reducing CLAI among GBMSM.

**Online interventions.** The majority (seven) of the nine studies that used online interventions were conducted in the USA [23, 42–45, 50, 51] and the other two were conducted in Taiwan [55] and Hong Kong [54] (Table 1 (colour purple in Figs 2–5)). The studies delivered cognitive or behavioural interventions online using video-games [42, 51], interactive modules, forums or apps [23, 44, 45, 55], short videos [54], Information-Motivation-Behavioural Skills (IMB) [50] or online chat with a live facilitator [43]. The majority (six) of studies aimed to reduce CLAI among HIV negative or undiagnosed GBMSM [23, 42–44, 54, 55] using self-reported outcomes, and three studies aimed to reduce HIV transmission risk behaviours in HIV diagnosed GBMSM [45, 50, 51]. Of the three studies that recruited HIV diagnosed GBMSM, one used cumulative STI incidence as the primary outcome measure [45] and the other two used reduced sexual risk behaviour (i.e. CLAI) [50, 51]. Two studies that recruited HIV diagnosed GBMSM were of high-quality [45, 50] and one was rated as mixed quality [51]. The first high quality study compared enhanced online internet-delivered tailored messages about safer-sex, disclosure and ART intervention against monthly sexual behaviour surveys, but did not find a difference in 12-month STI incidence (SES 0.17 95%CI -0.21, 0.55) (Fig 3) [45]. The second compared a four-session online HIV sexual risk reduction intervention (HINTS) using an IMB model with a time-matched 'healthy living' comparison [50], however the results after six months follow-up did not demonstrate a reduction in total number of CLAI acts (SES -0.15 95%CI -0.3, 0.001) (IRR: 0.91 95%CI 0.63, 1.30) and in fact showed a small but significant increase in the number of partners reported (SES 0.25 95%CI 0.09, 0.4) (IRR: 1.81 95% CI: 1.23, 2.68) [50]. The only other high quality RCT that used an online intervention tested the effect of a bespoke seven module intervention taking two hours, based on the information-motivation-behavioural skills model of HIV risk behaviour change [44]. Although the follow-up period for this intervention was very short (12 weeks), those in the intervention arm reported 44% lower prevalence of CLAI compared with controls (SES -0.27 95%CI -0.46 to -0.07) (Fig 4) [44]. The remaining RCTs were rated as mixed [23, 42, 43, 51, 54] or poor quality [55] (Table 2) and only one showed any intervention effect of increasing consistent condom use among GBMSM [55] (Fig 4).

**Contingency management for substance use.** The two Contingency Management (CM) RCTs were conducted among methamphetamine-using, GBMSM in the USA [46, 47] (Table 1 (colour grey in Figs 2–5)). One study was aimed at reducing substance use as well as CLAI among HIV negative men and also assessed increasing PEP initiation and course completion [46]. This study was rated as mixed quality and found no significant difference in CLAI at six months (SES 0.15 95%CI -0.03, 0.34) (Fig 4) [46].

The other study recruited substance using homeless GBMSM and assessed the impact of two culturally sensitive intervention programs on reduction of drug use and sexual partner numbers [47]. This study was rated poor quality for several reasons (see Table 2) including inappropriate control conditions and poor statistical analysis [47] and the intervention had no impact on reducing partner numbers (SES -0.14 95%CI -0.48, 0.19) (Fig 5).

**HIV pre-exposure prophylaxis.** This review included three high quality RCTs (five papers) among GBMSM that included PrEP in the experimental arm [49, 52, 57–59] and one mixed quality RCT (Table 2) [48]. Two studies were conducted in the USA [48, 49], one in the UK [52] and one in France and Canada [57] (Table 1 (colour red in Figs 2–5)). Two of the five papers reported on HIV incidence [52, 57], three on STI incidence [52, 58, 59], three on measures of condom use or CLAI [48, 49, 52] and one on partner numbers [52], one study reported on all four outcomes [52].

 

Two studies assessed the effect of PrEP on incident HIV infection and were both scored as high quality [52, 57]. The UK based open-label RCT, PROUD study (Pre-exposure prophylaxis to prevent the acquisition of HIV-1 infection) randomised participants to either immediate daily PrEP initiation or initiation after a one-year deferral and was the only study that provided data on all four chosen outcomes. An 86% (90% CI 64–96) proportionate reduction in HIV incidence was demonstrated (SES -1.06 95%CI -2.02, -0.41) (Fig 2) [52]. There was no significant difference in STI diagnosis between the two groups after 2 years of follow-up (SES 0.09 95%CI -0.03, 0.21) (Fig 3) or in partner numbers (10 or more Vs less than 10) (SES 0.11 95%CI -0.1, 0.33) at 2 years follow-up (Fig 4), although the intervention was not aimed at achieving these outcomes. However, a significantly larger proportion of men in the immediate PrEP group reported receptive CLAI with ten or more partners at 2 years follow-up (21% vs 12% p = 0.03) (SES 0.35 95%CI 0.05, 0.64) (Fig 5).

The multicentre RCT of high-risk GBMSM in Canada and France known as IPERGAY (Intervention Preventive de l'Exposition aux Risques avec et pour les Gays) reported a relative reduction in HIV incidence in the intervention group who received on demand PrEP of 86% (95%CI 40–98 p = 0.002) (SES -1.07 95%CI -2.33, -0.28) (Fig 2). Unlike previous PrEP trials where participants took daily PrEP, those in the IPERGAY trial took a loading dose of two pills 2–24 hours before sex, followed by daily pills for 48 hours after sex [57]. A separate study of participants from the ANRS IPERGAY study evaluated the impact of on-demand PrEP on herpes simplex virus (HSV)-1/2 incidence and demonstrated that oral PrEP failed to reduce incidence of HSV 1/2 in this population (SES (for HSV1) -0.26 95%CI:-1.05, 0.55. SES (for HSV2) -0.6 95%CI:-0.06, 0.14) (Fig 3) [59]. A sub-study of the ANRS IPERGAY study randomised consented participants to take, in addition to PrEP, either a single oral dose of 200 mg doxycycline PEP within 24 h after sex or no prophylaxis [58]. This was the only study to demonstrate a reduction in incident STI and reported that the occurrence of a first STI in participants taking PEP was lower than in those not taking PEP (HR 0·53; 95% CI 0·33–0·85; p = 0·008) (SES 0.5 95%CI 0.29, 0.88) (Fig 3) [58].

The mixed quality US-based RCT examined the effect of a nurse delivered cognitive behavioural intervention, compared to time and session matched nurse-led counselling control, on CLAI and PrEP adherence [48]. The study demonstrated no significant difference in adherence compared to information and support counselling control and no impact on CLAI (SES -0.03 95%CI: -0.21, 0.14) (Table 2) (Fig 4).

## Discussion

This systematic review of the literature examines recent evidence about the effectiveness of interventions to reduce HIV incidence among GBMSM in high income countries, and adds to the review conducted by Stromdahl et al in the period before 2013 [16]. Thirty-seven randomised controlled trials of five intervention types reported by thirty-nine papers were included in this review. Overall, PrEP was the only intervention that demonstrated a significant reduction in HIV incidence; in fact both high quality RCTs reported a reduction of 86% in HIV incidence [52, 57]. The PROUD study was conducted in the UK [52] and the ANRS IPERGAY study in France and Canada [57]. Prior to these two studies, iPREX, a large international study conducted in 2007 to 2009 in Peru, Ecuador, Thailand, Brazil, USA and South Africa (included in Stromdahl's review [16] but not included in this review due to data being collected before 2008 and no disaggregation of data from low and high countries) also demonstrated that PrEP provides significant protection against HIV acquisition [60]. As a result, PrEP has become a central part of HIV prevention interventions around the world and PrEP initiatives are currently offered in 78 countries [61].

The other interventions included in this review yielded mixed results and used other measures reflecting HIV risk such as self-reported CLAI or acquisition of STIs or partner numbers rather than HIV incidence. In terms of reducing risk of HIV transmission through STI management [62], PrEP combined with the antibiotic doxycycline prophylaxis was shown to reduce the occurrence of a first episode of bacterial STI compared to PrEP alone [58], however none of the other interventions had any impact on reducing STI incidence and the majority did not have any impact on partner numbers. In fact, only two interventions, one-to-one counselling [26] and a group intervention [36], demonstrated a small reduction in partner numbers. One-to-one counselling demonstrated some significant reductions in measures of CLAI [26, 29, 30, 31, 32, 35] among certain sub-populations such as newly diagnosed GBMSM [35] or substance using GBMSM [31] and an increase in consistent condom use among young, black GBMSM [30]. However, two other studies that used one to one counselling as an intervention showed either no significant reduction in our standardised effect size [28], or (in the chosen outcome of CLAI with casual partners) even an increase in CLAI [33]. Two group interventions demonstrated an increase in measures of condom use among Latino GBMSM [39, 40] and one group intervention demonstrated a small but significant reduction in measures of CLAI among substance using GBMSM [38]. A previous systematic review of behavioural interventions for Latino GBMSM also identified some successful interventions for this particular group but highlighted the need to better incorporate and describe cultural features if such interventions are to be successful [63]. There was some evidence that online interventions effectively reduced CLAI in both HIV diagnosed [44] and HIV negative MSM [42, 43] however the majority of follow-up times for these studies were relatively short ($</ = 6$ months).

It is important that the results of this systematic review are interpreted in the context of the restrictions placed upon it by only including randomised controlled trials, and the calendar years of included studies (2013–2021). This review retrieved far fewer intervention types than previous reviews [16, 64], possibly because the selection criteria restricted the study type to randomised controlled trials, however randomized controlled trials, when feasible, do provide the best evidence to assess the efficacy of interventions. In their review that did not restrict to RCTs, Stromdahl et al, (2015) strongly recommended four interventions for implementation in Europe: condom use, peer out-reach (providing information and peer support), peer-led group interventions (interactive group activities where a trained peer facilitates promotion of precautionary behaviours for HIV) and using universal coverage of ART and treatment as prevention (TasP). Our search did not retrieve any studies that investigated universal coverage of ART or TasP, probably because overwhelming earlier evidence of the efficacy of these interventions has reduced the need for studies examining the interventions' effectiveness over the past decade. Additionally, results from the PARTNER2 study demonstrated that the risk of sexual transmission of HIV in the context of virally suppressive ART in serodifferent gay partnerships is zero [65, 66] and the resulting Undetectable = Untransmittable campaign is championed by all major global health organisations (including WHO) and over one thousand community partners in over one hundred countries [67].

Despite the increasing number of countries adopting PrEP as a prevention intervention, ongoing HIV transmission remains in sub-populations of GBMSM, particularly those who are younger or from minority ethnic backgrounds, or who are recent migrants or are gender diverse [68, 69]. It has become increasingly clear that combination prevention that match the needs of a country or community, is necessary to end HIV transmission [70, 71]. Whilst the results from this systematic review (focusing on the evidence published between 2013 and 2021) suggest that PrEP as a biomedical intervention provides the strongest evidence for reducing HIV incidence, other interventions, outside the restrictions of this review, such as Treatment as Prevention, and rapid linkage to care following diagnosis and support to attain

viral suppression, have had a significant impact on HIV incidence [72]. Our results further demonstrate that targeted interventions such as online and group interventions, which can be tailored for individual communities, could also impact on sexual risk behaviours. However more high quality, culturally tailored and robust trials are needed. It is also increasingly understood that individual health behaviours are shaped by cultural contexts and social interactions, and that drivers of HIV transmission, as with many infections, are based on unmet need and social inequality which must be addressed as a cohesive approach to HIV prevention. Given the overwhelming evidence of the effectiveness of PrEP, more research is needed into the access and uptake of PrEP among populations that are not accessing it.

## Limitations of the data

Firstly, only studies that met the inclusion criteria for this systematic review were analysed. In particular we restricted to RCTs as the strongest study design for providing evidence on the effectiveness of an intervention. We excluded observational and non-randomised experimental studies, and therefore may have excluded potential prevention interventions supported by a lower level of evidence. Secondly, most of the studies relied on self-report data about changes in sexual behaviour, which are subject to social desirability and other biases, rather than collecting data on STI or HIV incidence. Studies which included both a self-reported outcome and an outcome of serological testing for STI reported mixed results [30, 34, 45]. In these cases, the self-report sexual behaviour data suggested the intervention had an effect but there was no difference in incident STIs between intervention and control arms. It is possible that studies that use self-report cannot reliably inform the evidence-base about the efficacy of a given intervention, particularly in the context of unblinded trials, where a social desirability effect on the endpoint in the intervention group may be especially relevant. Additionally, follow-up times for some of the studies was limited (three months) and almost all the studies were conducted in the USA, making generalisability to non-USA populations uncertain. Finally, the intervention groupings are broad and, within the various groups, none of the interventions were exactly the same. Many of the interventions were bespoke and tailored to specific sub-groups of GBMSM and readers should assess the overall effectiveness of an intervention type with that in mind.

## Limitations of the review

Our review was limited to the English language. While it is likely that relevant peer-reviewed studies were published in English language journals, we are aware that some potentially relevant papers may have been excluded. By restricting the review to studies with data collected after 2008 and published after 2013, we have limited our ability to make a greater case for the strength of evidence of particular interventions. However, the HIV epidemic among GBMSM in high income countries is continually changing, and factors such as migration, ethnicity, socio-economic status and health policy also have an impact on patterns of HIV transmission. Whilst this review has demonstrated that certain interventions were effective in specific populations, it is important that interventions are culturally appropriate in their implementation if they are to be accessible for all. Finally, as noted above, the review was limited by study type and several observational studies that may have added to the evidence base were excluded. It should also be noted that just because an RCT has failed to find an intervention effective it does not mean it is not effective, just that it has not been demonstrated to be effective in an RCT. However, the benefit of including only RCTs is that this review has limited the impact of bias in the overall conclusions.

## Conclusion

Our systematic review of randomised controlled trials from 2013 to 2021 evaluated five intervention types, of which PrEP was the only intervention that was consistently reported to be effective in reducing HIV incidence. Other interventions such as one-to-one counselling, online and group interventions had some impact on reducing high risk sexual behaviour such as CLAI for sub-populations of GBMSM. A systematic review focusing on calendar years before 2013 demonstrated the importance of interventions such as condom use, universal coverage of ART or treatment as prevention and PEP. Our results highlight the role of PrEP in combination HIV prevention but also emphasise the importance of culturally competent, targeted interventions that are designed and tested robustly.

## Supporting information

**S1 Checklist. PRISMA 2020 checklist.**
(DOCX)

**S1 Appendix.**
(DOCX)

**S2 Appendix.**
(DOCX)

## Author Contributions

**Conceptualization:** Fiona C. Lampe, Andrew Phillips, Alison J. Rodger, Valentina Cambiano.

**Data curation:** Janey Sewell, Ibidun Fakoya, Fiona C. Lampe, Alison Howarth, Valentina Cambiano.

**Formal analysis:** Janey Sewell, Fiona C. Lampe.

**Funding acquisition:** Fiona C. Lampe, Andrew Phillips.

**Investigation:** Janey Sewell, Valentina Cambiano.

**Methodology:** Janey Sewell, Alison Howarth, Valentina Cambiano.

**Supervision:** Fiona C. Lampe, Alison J. Rodger, Valentina Cambiano.

**Visualization:** Janey Sewell, Fiona C. Lampe, Valentina Cambiano.

**Writing – original draft:** Janey Sewell.

**Writing – review & editing:** Janey Sewell, Ibidun Fakoya, Fiona C. Lampe, Alison Howarth, Andrew Phillips, Fiona Burns, Alison J. Rodger, Valentina Cambiano.

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
