## [Decision Letter · Decision Letter 0]

10 Aug 2022

PONE-D-22-12178Effectiveness of interventions aimed at reducing HIV acquisition and transmission among gay and bisexual men who have sex with men (GBMSM) in high income settings: a systematic review.PLOS ONE

Dear Dr. Sewell,

Thank you for submitting your manuscript to PLOS ONE. After careful consideration, we feel that it has merit but does not fully meet PLOS ONE’s publication criteria as it currently stands. Therefore, we invite you to submit a revised version of the manuscript that addresses the points raised during the review process.

As you will see, your manuscript was well received overall. I ask that you attend to all of the reviewer's comments as I think they make good points that could strengthen your manuscript.

We look forward to receiving your revised manuscript.

Kind regards,

Ethan Moitra

Academic Editor

PLOS ONE

Journal Requirements:

a) Did participants provide their written or verbal informed consent to participate in this study?

b) If consent was verbal, please explain i) why written consent was not obtained, ii) how you documented participant consent, and iii) whether the ethics committees/IRB approved this consent procedure

"This research was supported by UK Research and Innovation through the Centre for Research into Energy Demand Solutions (grant number EP/R035288/). The funders have/had no role in study design, data collection and analysis, decision to publish or preparation of the manuscript.  Prof. Tadj Oreszczyn (UCL) supported the research through extensive discussions."

"This research was supported by UK Research and Innovation through the Centre for Research into Energy Demand Solutions (grant number EP/R035288/). The funders have/had no role in study design, data collection and analysis, decision to publish or preparation of the manuscript."

Reviewers' comments:

Reviewer's Responses to Questions

**Comments to the Author**

1. Is the manuscript technically sound, and do the data support the conclusions?

Reviewer #1: Yes

Reviewer #2: Partly

2. Has the statistical analysis been performed appropriately and rigorously? 

Reviewer #1: N/A

Reviewer #2: Yes

3. Have the authors made all data underlying the findings in their manuscript fully available?

Reviewer #1: Yes

Reviewer #2: Yes

4. Is the manuscript presented in an intelligible fashion and written in standard English?

Reviewer #1: Yes

Reviewer #2: Yes

5. Review Comments to the Author

Reviewer #1: The Authors present their systematic review paper well and in a layout that reads well. The methods are well described to explain the findings. The authors were able to assess for and be able to control foe bias in their inclusion methods. The references given are relevant to the subject and are adequate.

The tables and figures are well laid out and easy to read.

Reviewer #2: Thank you for inviting me to review this important paper. The paper aims to undertake a systematic review of prevention interventions in relation to reducing HIV transmission adding. This exercise was last undertaken in 2013. The paper is well written, relevant, clear and I have no issue with the statistical analyses.

However, I think the paper would benefit from placing the results in the wider context of HIV prevention.

General comments

Firstly some interventions lend themselves more easily to be studied under an RCT design, and while the necessity of restricting to these studies only is necessary for a quantitative systematic review, it does lead to a risk of not including evidence of other interventions that cannot be studies as well through this design. Similarly the evidence base for a lot of strong interventions TASP etc apparent before this time frame. While this acknowledged on page 36 para 1, I think the authors need to be clear in their conclusions their analysis is only related to what met their criteria, and that their findings do not replace the strong evidence of tASP etc.

Secondly, the paper would benefit to put the context of why this is now relevant – most high income countries have an ambition to end HIV transmission in gay and bisexual men and this work is needed to provide evidence of where efforts should be focussed

The intro and abstract correctly set the context of that a reduction in HIV transmission is not equally felt across all gay men, and ethnicity minorities etc are more likely tobe left behind. However, at present paper provides little insight of how the interventions assessed can help reduce inequalities. There is some mention of specific groups being under study in table 1, however, there is little synthesis of evidence about reducing inequalities other than stating specific interventions worked for some populations. It may not be possible to say much more the need that evidence based interventions that showed strongest effect need to be culturally competent in their implementation to make them accessible - but needs to be included as limitation in discussion.

There needs to be acknowledgement that a hierarchy of prevention interventions is not particularly helpful – it has been shown that combination prevention is necessary to end HIV transmission -focussing on PrEP only for instance, neglects the need to reduce transmission by ensuring rapid linkage to care following diagnosis and support to attain viral suppression.

Apologies if I missed it but it is not clear while this was restricted to RCTs only – presumably to allow the statistical analysis…

Major comments

Major comments:

Abstract - please put setting in introduction since this is not a universal fact

Abstract – conclusion suggest reword as “our systematic review of the recent evidence that we were able to analysedindicates” – I don’t think the design of the systematic review allowed for a comprehensive review of all prevention activities, e.g. including treatment as prevention, PEP, condoms, etc due to the time frame and necessity of being RCT design.

In intro on page 10 , last sentence of intro can you quickly explain why you are restricting to RCT design, when Stromdahl did not, given your aims are to identify prevention efficacy in the years since Stromdahl published.

Methods – page 10 last line – outcome is condomless anal intercourse, how can this be interpreted as an outcome related to HIV transmission when you are including PrEP as an intervention? Fir instance on page 15 one paper is cited as PrEP having a small increase in CLAI….. does that matter? Perhaps only restrict prep to the HIV incidence?

Page 30 last para – surely issue is lack of access to Prep in subpopulations of MSM – perhaps better to say need interventions and research to increase uptake in these populations given overwhelming evidence of effectiveness

Page 37 limitations of the data – only studies that met criteria analysed and these are likely underrepresent needs of groups with unmet needs

Data synthesis – does this mean that there was no studies of other interventions, or that those interventions did not meet the criteria (sufficient quality, RCT etc)

In results, perhaps make clear you are first looking at the impact on the outcomes, and then going through each intervention as it is hard to follow atm

Conclusion needs reworking to show applicability on sub populations of gay men, and to acknowledge this does not replace evidence of tasp ETC, and need for combination prevention.

6. PLOS authors have the option to publish the peer review history of their article (what does this mean?). If published, this will include your full peer review and any attached files.

Reviewer #1: **Yes: **Ubaldo Mushabe Bahemuka

Reviewer #2: **Yes: **Alison Brown

---

## [Author Response · Author response to Decision Letter 0]

16 Sep 2022

Journal requirements - Response to editor comments

1) We were asked to amend our current ethics statement to address the following concerns: a) Did participants provide their written or verbal informed consent to participate in this study? b) If consent was verbal, please explain i) why written consent was not obtained, ii) how you documented participant consent, and iii) whether the ethics committees/IRB approved this consent procedure.

Response: We have added an Ethics statement at the end of the manuscript that reads: “As this was a systematic review there were no participants involved in the writing or analysis of the paper.”

2) Funding: It was noted that we have provided funding information that is not currently declared in our Funding Statement. However, funding information should not appear in the Acknowledgments section or other areas of your manuscript. We will only publish funding information present in the Funding Statement section of the online submission form. Please remove any funding-related text from the manuscript and let us know how you would like to update your Funding Statement. "This research was supported by UK Research and Innovation through the Centre for Research into Energy Demand Solutions (grant number EP/R035288/). The funders have/had no role in study design, data collection and analysis, decision to publish or preparation of the manuscript. Prof. Tadj Oreszczyn (UCL) supported the research through extensive discussions."

Response: There is no Acknowledgements section in the manuscript that we submitted and the above statement does not appear in the manuscript. As per the advice above we have removed the funding statement from the first page of the manuscript and the funding statement has been included in the cover letter and should read: 

“This systematic review was undertaken as part of a programme of research which was funded by the NIHR under its Programme Grants for Applied Research Programme (Grant Reference Number RP-PG-1212-20006): A comprehensive assessment of the cost-effectiveness of HIV prevention and testing strategies, including HIV self-testing, among men who have sex with men (MSM) in the UK (PANTHEON). The views expressed are those of the authors and not necessarily those of the NIHR or the Department of Health and Social Care. This review was not registered. The review protocol and data extraction form can be accessed by contacting the corresponding author (j.sewell@ucl.ac.uk)”

Response to Reviewers' comments:

1) Firstly some interventions lend themselves more easily to be studied under an RCT design, and while the necessity of restricting to these studies only is necessary for a quantitative systematic review, it does lead to a risk of not including evidence of other interventions that cannot be studies as well through this design. Similarly the evidence base for a lot of strong interventions TASP etc apparent before this time frame. While this acknowledged on page 36 para 1, I think the authors need to be clear in their conclusions their analysis is only related to what met their criteria, and that their findings do not replace the strong evidence of tASP etc.

Response: Thank you for this relevant comment which we agree with. The following sentences (additional text in Italics) have been added to the Discussion, paragraph 2 and is also addressed in points 3 and 13:

“It is important that the results of this systematic review are interpreted in the context of the restrictions placed upon it by only including randomised controlled trials, and the calendar years of included studies (2013-2021). This review retrieved far fewer intervention types than previous reviews (16, 64), possibly because the selection criteria restricted the study type to randomised controlled trials, however randomized controlled trials, when feasible, do provide the best evidence to assess the efficacy of interventions. In their review that did not restrict to RCTs, Stromdahl et al…” 

2) Secondly, the paper would benefit to put the context of why this is now relevant – most high income countries have an ambition to end HIV transmission in gay and bisexual men and this work is needed to provide evidence of where efforts should be focussed. The intro and abstract correctly set the context of that a reduction in HIV transmission is not equally felt across all gay men, and ethnicity minorities etc are more likely to be left behind. However, at present paper provides little insight of how the interventions assessed can help reduce inequalities. There is some mention of specific groups being under study in table 1, however, there is little synthesis of evidence about reducing inequalities other than stating specific interventions worked for some populations. It may not be possible to say much more the need that evidence based interventions that showed strongest effect need to be culturally competent in their implementation to make them accessible - but needs to be included as limitation in discussion.

Response: Agreed, thank you for highlighting this. The following has been added to the limitations (additional text in italics):

“However, the HIV epidemic among GBMSM in high income countries is continually changing, and factors such as migration, ethnicity, socio-economic status and health policy also have an impact on patterns of HIV transmission. Whilst this review has demonstrated that certain interventions were effective in specific populations, it is important that interventions are culturally competent in their implementation if they are to be accessible for all.”

In addition “culturally competent” has been added to the recommendations for the planning of interventions in the overall conclusion and abstract.

3) There needs to be acknowledgement that a hierarchy of prevention interventions is not particularly helpful – it has been shown that combination prevention is necessary to end HIV transmission -focussing on PrEP only for instance, neglects the need to reduce transmission by ensuring rapid linkage to care following diagnosis and support to attain viral suppression.

Response: Thank you for pointing this out. The following has been added to the Discussion, paragraph 2 and the ‘limitations of the review’ section (additional text in Italics):

“It has become increasingly clear that combination prevention that match the needs of a country or community is necessary to end HIV transmission (70, 71 ). Whilst the results from this systematic review (focusing on the evidence published between 2013 and 2021) suggest that PrEP as a biomedical intervention provides the strongest evidence for reducing HIV incidence, other HIV prevention interventions, outside the restrictions of this review, such as Treatment as Prevention, and rapid linkage to care following diagnosis and support to attain viral suppression, have had a significant impact on HIV incidence (72). Our results further demonstrate that targeted interventions such as online and group interventions, which can be tailored for individual communities, could also impact on sexual risk behaviours. Finally, as noted above, the review was limited by study type and several observational studies that may have added to the evidence base were excluded. It should also be noted that just because an RCT has failed to find an intervention effective it does not mean it is not effective, just that it has not been demonstrated to be effective in an RCT.”

4) Apologies if I missed it but it is not clear while this was restricted to RCTs only – presumably to allow the statistical analysis. 

Response: The rationale for restricting to randomized controlled trials is that they are the gold standard to assess the efficacy of interventions. They are the only type of study able to establish causation as randomization minimizes confounding and if the trial is sufficiently large it ensures that the groups are comparable (even for unobserved characteristics) apart from the intervention of interest. However, we do recognize that RCTs have some limitations, including having an highly controlled setting, which may not always mimic real life situations. In addition, there are circumstances for which some interventions may not be evaluated because of feasibility or ethical concerns and so for some interventions we might have not found evidence as they can’t be evaluated in an RCT.

5) Abstract - please put setting in introduction since this is not a universal fact

Response: Additional text (in Italics) added to Background:

“We aimed to identify and describe recent studies evaluating the effectiveness of HIV prevention interventions for GBMSM in high income countries”

6) Abstract – conclusion suggest reword as “our systematic review of the recent evidence that we were able to analyse indicates” – I don’t think the design of the systematic review allowed for a comprehensive review of all prevention activities, e.g. including treatment as prevention, PEP, condoms, etc due to the time frame and necessity of being RCT design.

Response: Additional text (in Italics) added as suggested to Conclusion:

“Our systematic review of the recent evidence that we were able to analyse indicates that PrEP is the most effective intervention for reducing HIV acquisition among GBMSM. Targeted and culturally competent behavioural interventions for sub-populations of GBMSM vulnerable to HIV infection and other STIs could also be considered particularly for GBMSM who cannot access or decline to use PrEP.”

7) In intro on page 10 , last sentence of intro can you quickly explain why you are restricting to RCT design, when Stromdahl did not, given your aims are to identify prevention efficacy in the years since Stromdahl published.

Response: Thank you, the additional text in Italics has been added:

“In contrast to Stromdahl et al’s review we restricted the studies included in this review to randomised controlled trials, as these are the gold standard to establish the effectiveness of an intervention, as confounding is minimised due to randomization.” 

8)Methods – page 10 last line – outcome is condomless anal intercourse, how can this be interpreted as an outcome related to HIV transmission when you are including PrEP as an intervention? For instance on page 15 one paper is cited as PrEP having a small increase in CLAI….. does that matter? Perhaps only restrict prep to the HIV incidence?

Response: Thank you for pointing this out. We thought it was important to recognise and report findings on all four of our chosen outcomes, if reported, particularly as there is conflicting evidence on the effect that PrEP actually has on CLAI. We agree that for a person adherent to PrEP an increase in CLAI does not actually have an impact on the risk of HIV acquisition however we think it is important to still report it.

9) Page 30 last para – surely issue is lack of access to Prep in subpopulations of MSM – perhaps better to say need interventions and research to increase uptake in these populations given overwhelming evidence of effectiveness

Response: We agree. The following sentence (additional text in Italics) has been added: 

“Given the overwhelming evidence of the effectiveness of PrEP, more research is needed into strategies to increase the uptake of PrEP among populations that are not accessing it.”

10)Page 37 limitations of the data – only studies that met criteria analysed and these are likely underrepresent needs of groups with unmet needs

Response: Thank you for pointing this out. The following sentences have been added to the limitations of the data:

“Firstly, only studies that met the inclusion criteria for this systematic review were analysed. In particular we restricted to RCTs as the strongest study design for providing evidence on the effectiveness of an intervention. We excluded observational and non-randomised experimental studies, and therefore may have excluded potential prevention interventions supported by a lower level of evidence." 

11) Data synthesis – does this mean that there was no studies of other interventions, or that those interventions did not meet the criteria (sufficient quality, RCT etc)

Response: Yes, it means that there were no other RCT studies that were identified by our searches. We did not exclude on quality.

12) In results, perhaps make clear you are first looking at the impact on the outcomes, and then going through each intervention as it is hard to follow atm

Response: Thank you, we have added two subheadings in the Results to make this clearer:

1) Impact of the intervention on the four chosen outcomes

2) Intervention assessment

13) Conclusion needs reworking to show applicability on sub populations of gay men, and to acknowledge this does not replace evidence of tasp ETC, and need for combination prevention.

Response: Thank you for pointing this out, we have acknowledged that the results do not replace evidence of TasP in response to previous comments (response 1) and we have reworked the conclusion as follows (additional text in Italics):

“Our systematic review of randomised controlled trials from 2013 to 2021 evaluated five intervention types, of which PrEP was the only intervention that was consistently reported to be effective in reducing HIV incidence. Other interventions such as one-to-one counselling, online and group interventions had some impact on reducing high risk sexual behaviour such as CLAI for sub-populations of GBMSM. 

A systematic review focusing on calendar years before 2013 demonstrated the importance of interventions such as condom use, universal coverage of ART or treatment as prevention and PEP. Our results highlight the role of PrEP in combination HIV prevention but also emphasise the importance of culturally competent, targeted interventions that are designed and tested robustly.”

---

## [Decision Letter · Decision Letter 1]

2 Oct 2022

Effectiveness of interventions aimed at reducing HIV acquisition and transmission among gay and bisexual men who have sex with men (GBMSM) in high income settings: a systematic review.

PONE-D-22-12178R1

Dear Dr. Sewell,

We’re pleased to inform you that your manuscript has been judged scientifically suitable for publication and will be formally accepted for publication once it meets all outstanding technical requirements. I was unable to re-engage one of the prior reviewers; as such, I assumed the reviewer's role and judged that you adequately responded to their critiques. Nice job!

Kind regards,

Ethan Moitra

Brown University

Academic Editor

PLOS ONE

Additional Editor Comments (optional):

Reviewers' comments:

Reviewer's Responses to Questions

**Comments to the Author**

1. If the authors have adequately addressed your comments raised in a previous round of review and you feel that this manuscript is now acceptable for publication, you may indicate that here to bypass the “Comments to the Author” section, enter your conflict of interest statement in the “Confidential to Editor” section, and submit your "Accept" recommendation.

Reviewer #1: All comments have been addressed

2. Is the manuscript technically sound, and do the data support the conclusions?

Reviewer #1: Yes

3. Has the statistical analysis been performed appropriately and rigorously? 

Reviewer #1: Yes

4. Have the authors made all data underlying the findings in their manuscript fully available?

Reviewer #1: Yes

5. Is the manuscript presented in an intelligible fashion and written in standard English?

Reviewer #1: Yes

6. Review Comments to the Author

Reviewer #1: (No Response)

7. PLOS authors have the option to publish the peer review history of their article (what does this mean?). If published, this will include your full peer review and any attached files.

Reviewer #1: **Yes: **Ubaldo Mushabe Bahemuka

---

## [Editor Report · Acceptance letter]

6 Oct 2022

PONE-D-22-12178R1 

Effectiveness of interventions aimed at reducing HIV acquisition and transmission among gay and bisexual men who have sex with men (GBMSM) in high income settings: a systematic review. 

Dear Dr. Sewell:

I'm pleased to inform you that your manuscript has been deemed suitable for publication in PLOS ONE. Congratulations! Your manuscript is now with our production department. 

Kind regards, 

on behalf of

Dr. Ethan Moitra 

Academic Editor

PLOS ONE